# The fragmentation-induced fluidisation of pyroclastic density currents

Eric C. P. Breard [1,2] ✉, Josef Dufek[2], Sylvain Charbonnier[3], Valentin Gueugneau[3], Thomas Giachetti[2] & Braden Walsh[4]

Pyroclastic density currents (PDCs) are the most lethal volcanic process on Earth. Forecasting their inundation area is essential to mitigate their risk, but existing models are limited by our poor understanding of their dynamics. Here, we explore the role of evolving grain-size distribution in controlling the runout of the most common PDCs, known as block-and-ash flows (BAFs). Through a combination of theory, analysis of deposits and experiments of natural mixtures, we show that rapid changes of the grain-size distribution transported in BAFs result in the reduction of pore volume (compaction) within the first kilometres of their runout. We then use a multiphase flow model to show how the compressibility of granular mixtures leads to fragmentation-induced fluidisation (FIF) and excess pore-fluid pressure in BAFs. This process dominates the first ~2 km of their runout, where the effective friction coefficient is progressively reduced. Beyond that distance, transport is modulated by diffusion of the excess pore pressure. Fragmentation-induced fluidisation provides a physical basis to explain the decades-long use of low effective friction coefficients used in depth-averaged simulations required to match observed flow inundation.

Fatalities from volcanic eruptions in the past decades have been largely related to pyroclastic density currents (PDCs)[1]. The most common PDCs are known as block-and-ash flows[2–4] (BAFs), which form by gravitational collapse of a hot lava dome[5,6] lava lakes or flows[7] or perched pyroclastic debris[8], and have been the deadliest volcanic processes in the past century[1,9]. Observations and depositional evidence suggest these flows are characterised by a concentrated basal avalanche (underflow) overlain by a dilute turbulent ash-cloud (ash-cloud surge)[10–13]. Nevertheless, key properties of these layers remain obscured, particularly in the concentrated underflow that dominates the mass and momentum budget of these flows[14]. The extreme mobility of BAFs defies our understanding of the dynamics of granular flows and prevents effective risk mitigation[15]. Their long runouts are particularly confounding as they initiate with relatively low energy compared to PDCs fed by column collapse[16–18].

Considerable efforts by hazard practitioners and numerical modellers have been made to forecast the propagation of PDCs by using depth-averaged models on 3D topography[10,15,19–25]. These studies suggest the existence of a mechanism that reduces the effective friction of the volcanic mixtures that are otherwise highly frictional when static (angle of repose of ~30–35°[26]). Lowering the effective friction could be achieved through the generation of elevated-pore pressure[26–29], which makes the mixture fluid-like (fluidised). However, the source of the fluidisation mechanism in BAFs remains elusive because, unlike column collapse PDC, they do not form by impact of a column collapse or rapid-sedimentation, and are coarse (permeable) mixtures in proximal reaches[30]. For example, the 2018 Fuego dome-less BAFs that originated from the collapse (>70% volume[31]) of existing "perched" proximal tephra, lava, and spatter were some of the most mobile and deadly events of the past century[8,32], and yet cannot be explained by our existing knowledge of auto-fluidisation[26,33].

[1]School of Geosciences, University of Edinburgh, Edinburgh, UK. [2]Department of Earth Sciences, University of Oregon, Eugene, OR, USA. [3]School of Geosciences, University of South Florida, Tampa, FL, USA. [4]Centre for Natural Hazards Research, Department of Earth Sciences, Simon Fraser University, Burnaby, BC, Canada. ✉e-mail: Eric.Breard@ed.ac.uk

The evolving grain-size distribution (GSD) of block-and-ash flows may provide such a mechanism. In column collapse PDCs, secondary fragmentation, or comminution, has led to measurable rounding differences and is linked to enhanced runout distances[34–37]. BAF deposits start as a mixture dominated by blocks (>64 mm) and transform into a blend of blocks immersed in fines-rich (<4 mm) matrix (Fig. 1a). However, the feedback between the rapid evolution of the GSD in BAFs prior to proximal deposition, evolving permeability (i.e. abundance of fines and porosity[38–40]), and the subsequent role of the interstitial gas modifying the effective friction of the mixture has never been studied.

In this work, we introduce the mechanism of fragmentation-induced fluidisation (FIF), in which gas-particle coupling in an evolving permeable mixture generates elevated pore pressure in the granular mixture. As demonstrated in experimental studies, the permeability and pore-pressure diffusion timescales are controlled by the polydispersity. A key component of the fragmentation-induced fluidisation process is a reduction in the void space through the production of fine particles via cascading fragmentation. As a simple thought experiment, one would expect that this reduction in void space due to fragmentation would increase the pore pressure at the same time as reducing permeability and the ability of the flow to expel trapped gas. To better understand how such a process plays a role in the formation and propagation of BAFs and explain their high mobility, we first analysed BAF events from the well-studied eruptions of Merapi volcano (Indonesia) that has repeatedly produced long-runout flows, causing fatalities.

## Results and discussion
### The grain-size distribution of BAFs and self-limiting fragmentation

The three BAF events of June 14, 2006 (runout of ~7 km), October 26, 2010 (runout of ~7.5 km) and November 5, 2010 (runout of ~16 km) at Merapi were triggered by different eruptive processes[41–43]. Yet, their valley-filling deposits share strong similarities (Fig. 1b). We arbitrarily define proximal, medial, and distal regions by splitting the runout equally in three. The GSDs are bimodal (Fig. 1b), coarse, very poorly sorted, and show little change in medial to distal locations (final 2/3rd

of their runout, Fig. 1c). The lack of strong changes in GSDs past medial distances implies little segregation or secondary fragmentation occurred late in the flow and allows us to explore the processes that initiated it. The particle size distribution of rocks that undergo self-similar fragmentation follows a power law $N = \lambda d^{-D}$, where $N$ is the number of particles greater than size $d$, $\lambda$ is a scaling factor, and $D$ is the power-law exponent (fractal dimension[44]). The GSDs of the three Merapi events studied follow a power-law size distribution with $D = 3.0 \pm 0.1$ for most samples (Fig. 1d), which is within the range reported for all PDCs of 2.9–3.4[45,46] and analogous to rock avalanches[47]. Since all GSDs studied here follow comparable trends whereas the dome collapse initiation mechanisms differ between events[41,43], fragmentation of clasts in BAFs likely follows a self-similar process driven by attrition where particles break from either impact, compression, frictional or shear forces (and combination of these).

The fragmentation of clasts by attrition appears to be self-limiting since the proximal deposits are systematically matrix supported and show little change with distance. BAFs typically have 30–50 vol.% of their solid mixture represented by the matrix (typically <~4 mm[43]). When granular flows have <~30 vol.% of fines, the mixture is clast supported and the contribution of the stress at contacts is carried primarily by the coarse fractions[48,49]. However, the contributions are reversed when fines make >~30 vol.% of the mixture (matrix supported deposit) to an extent where the contact stresses in the coarse fractions are <10% of the total stress when fines reach ~40 vol.%[49]. These findings are independent of the size ratios between the coarse and fine modes for a size ratio >5[49]. Since the size ratio between the modes is on the order of $10^2$ in PDCs[50,51], we infer the self-limiting attrition process is due to the lower frequency of impacts between coarse particles and the small contact stresses during these impacts, which limit further breakage. As a result, this self-limiting process prevents the transformation of (bimodal) block-and-ash flows into (unimodal) ash-dominated flows.

### The packing of polydisperse grain-size distributions
In the absence of an external gas source, fluidisation of a granular mixture immersed in a fluid can result from changes in packing (i.e.,

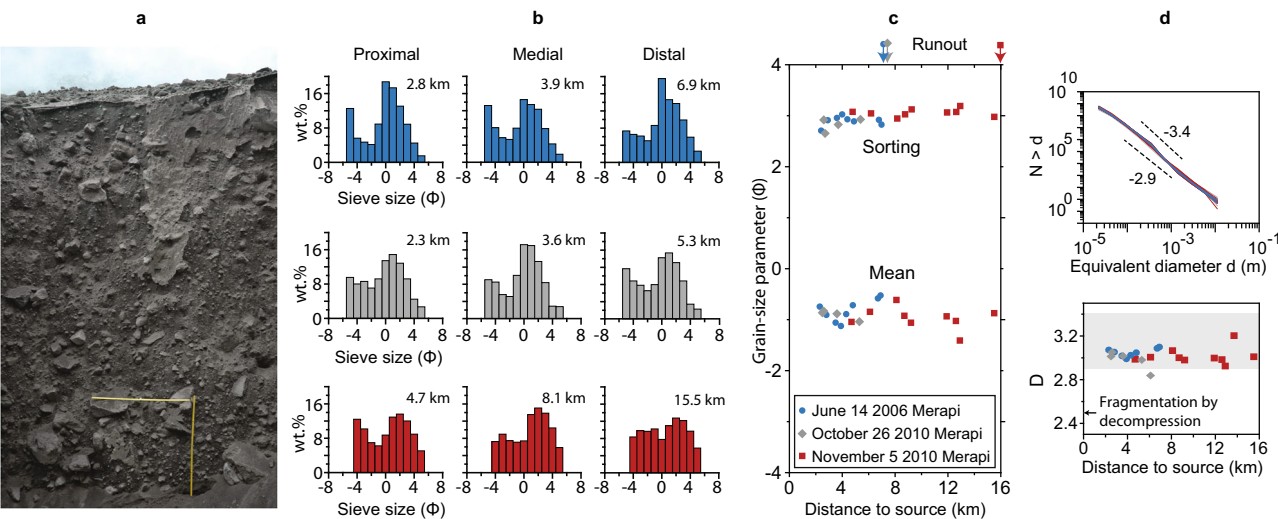

**Fig. 1 | Block-and-ash flow deposits at Merapi volcano. a** Outcrop of the block-and-ash flow deposit from the June 14, 2006, event at Merapi volcano at 2.8 km from source. Each yellow bar is 1 m long. **b** Proximal, medial and distal grain-size distributions of samples collected at Merapi for all three BAFs of June 14, 2006 (blue), October 26, 2010 (grey) and November 5, 2010 (red) ($\Phi = -\log_2(d[\text{mm}])$). Note that samples in the "proximal" region were collected close to the proximal/media transition. **c** Sorting and mean grain-size parameters for all samples plotted with distance. The runout of each event is indicated on the top with vertical arrows.

**d** Number of particles with diameter >$d$ as a function of particle size for all samples of the three BAF events at Merapi. The two black reference dashed lines illustrate the difference in slope between power-law distributions with fractal dimension of 2.9 and 3.4[46], which are typical bounds of the slope measured in other PDC deposit samples around the world. The power-law exponent $D$ is plotted against distance for all samples. The greyed box represents the range of $D$ exponents measured in other PDC deposits.

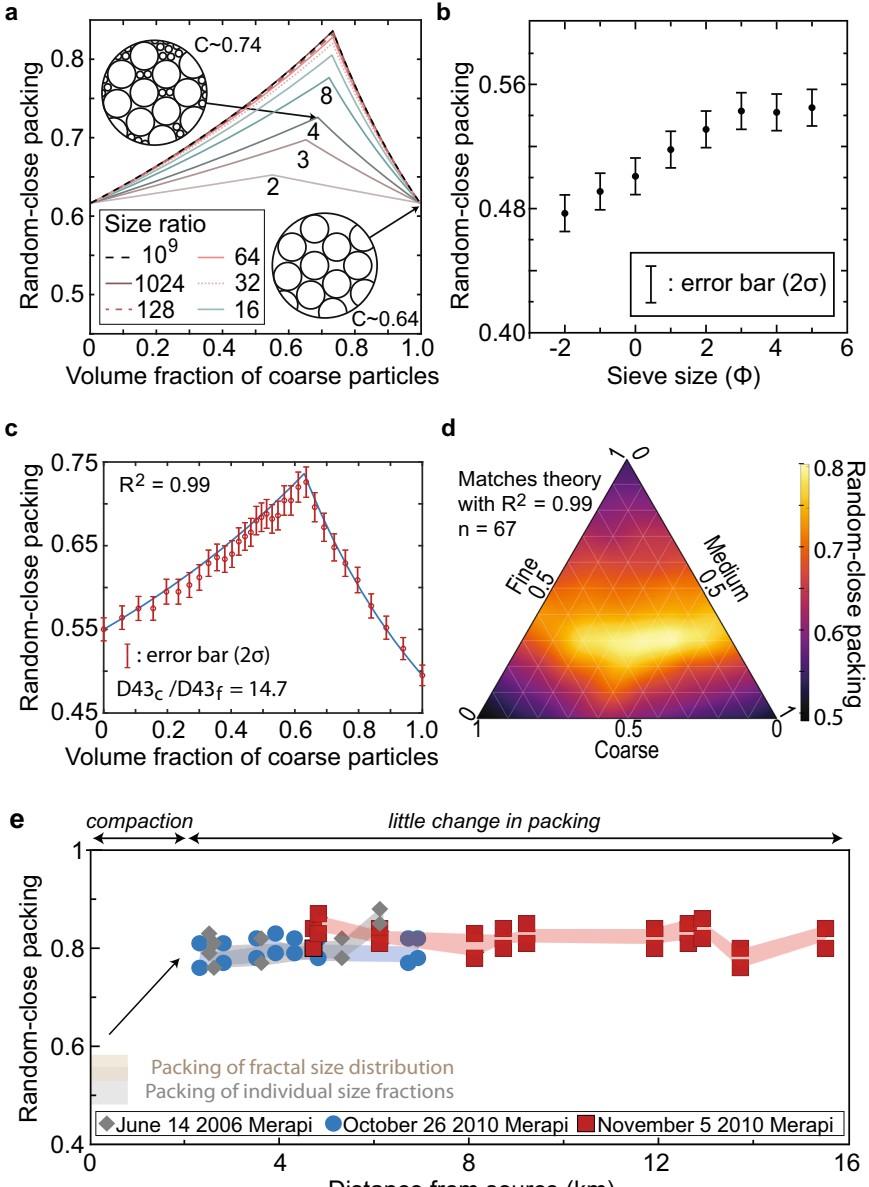

**Fig. 2 | The packing of polydisperse volcanic mixtures. a** Theoretical prediction of the random-close packing of bidisperse spheres with various size ratios shown in legend and labelled on curves[54]. **b** Random-close packing (black filled circles) of various sieved size fractions of BAF material with error bars representing the standard deviation in measurements (2σ). **c** Experimental measurements of random-close packing of BAF material with D43 size ratio between the coarse and fine fractions D43$_c$/D43$_f$ = 14.7 (red circles) and theoretical prediction (blue line) based on the packing of the two endmembers using Yu and Standish[54]. **d** Experimental measurements of the packing of tridisperse distributions made of BAF material with size ratios D43$_c$/D43$_f$ = 14.7, D43$_m$/D43$_f$ = 3.6 and D43$_c$/D43$_m$ = 4.1. **e** Theoretical predictions of the random-close packing of all BAF samples for the three events at Merapi volcano, plotted against distance. The grey diamond, blue circle and red squares are derived using the spherepack1D algorithm[55]. The light blue, red and grey areas are derived by using the theory from Maroof et al.[68]. The light tan box represents the range of random-close packing fraction of particles in 1Φ size bins and the grey box shows the range of random-close packing of a mixture with fractal size distribution with fractal dimension = 1.5.

compaction[52]), which occurs as a response to the evolution of the GSD with time (i.e., making of fines filling void space). The maximum particle volumetric concentration of monodisperse (i.e., single size) spheres randomly packed is ~0.637[53] and independent of particle size (also called random-close packing[53]). However, the random-close packing of bidisperse spherical mixtures is a function of both the size ratio and the proportion of the two solid populations[54] (Fig. 2a). The packing of volcanic mixtures is assumed to be equivalent to spheres' and constant regardless of the grain-size transported[20,36]. Here we challenge this assumption since the random-close packing of the distinct size fractions comprising the fines fraction of BAFs is much lower than spheres', with value spanning 0.48–0.54 (Fig. 2b). The lower

packing values compared to spheres are due to the irregular particle shapes with a circularity of 0.895–0.938 and roundness of 0.458–0.585, which appear to be size dependent (Supplementary Fig. 1a, b). Once blended, the measured random-close packing of bimodal distributions of BAF material is compared to the theoretical prediction made using the law for bidisperse mixtures, knowing a priori the packing of the two endmembers and the size ratio using the ratio of particle volume diameter of each distribution (D43$_{coarse}$/D43$_{fine}$). The excellent match ($R^2$ = 0.99) between measurements and theoretical predictions for the bimodal distribution, with size ratios of 14.7 (Fig. 2c) and 3.6 (Supplementary Fig. 1c), holds for the polynomial size distribution (Fig. 2d), validating the method used and suggesting

that existing theoretical predictions for spheres[55] can be adapted to predict the packing of complex volcanic mixtures.

Natural GSDs are often log-normal size distributions, whose packing is strongly controlled by the sorting (Supplementary Fig. 1d). This exemplifies the need to consider the GSD when estimating the packing of a granular mixture. However, measuring the packing of BAF deposits in situ was not feasible, nor was recreating the close-packing of the mixture in the laboratory due to segregation of the size fractions during transport. Therefore, we use existing theoretical work on the packing of polydisperse spherical and non-spherical size distributions to derive two independent theoretical predictions of the random-close packing from BAF deposit samples collected at Merapi (see 'Methods'). These packing results are plotted with distance and show that random-close packing values of all three Merapi BAF events are within $0.80 \pm 0.05$ (Fig. 2e). Since the sampling of BAF deposits did not capture the largest size fractions (>64 mm), which would increase the measured sorting grain-size, the random-close packing reported are minimum. Nevertheless, these values are much larger than assumed values of 0.6 used in various PDC models. While the initial grain-size distribution of the avalanche is not measurable, the random-close packing of the initial mixture is estimated to be ~0.53–0.585 ('Methods'), once the blocks collapsed gravitationally and fragment due to the presence of internal stresses (Fractal dimension $D$-1.5[56]). The development of the granular avalanche on the steep slope (>35°) transforms the GSD to reach a fractal dimensions $D$-3 prior to ~2.5 km of runout (i.e., most proximal sampling location), which subsequently modifies the maximum packing of the mixture. We now investigate the interplay between the evolving grain-size distribution, permeability, maximum packing and effective friction in these flows.

## The fragmentation-induced fluidisation (FIF) mechanism

All existing depth-averaged models assume a constant grain-size and packing with distance, but these two-phase flows (solid and gas) are highly time-variant systems where changes of the packing will impact the interstitial pore-fluid pressure and subsequently change the effective frictional stress. To examine the feedback between the transformation of the grain-size, subsequent change of the maximum packing, and the interstitial pore-fluid pressure, we employed a multiphase numerical model using an Eulerian–Eulerian framework ('Methods'). We simulate the evolution of a static vertical column made of hot compressible gas (air) and particles at 500 °C[57,58] in 2D, and use the June 14, 2006 event at Merapi as a reference for the physical properties of the mixture, which are better constrained than the other two events thanks to the high resolution sampling of its deposits[10,39]. The granular column is initiated as a mixture of particles of $10^{-2}$ m in diameter representing the coarse mode that fragments into a fraction of fines of $10^{-4}$ m in diameter. Since the mechanics of the fragmentation process are unknown, we imposed rates of fragmentation of the coarse solid phase present spanning $55 \times \text{Fraction}_{\text{coarse}}$ to $1 \times \text{Fraction}_{\text{coarse}}$ kg m$^{-3}$ s$^{-1}$. These values are motivated by flow durations and realistic range of timescales from 10 s to 10 min needed to create the mass of ash per unit volume observed in the BAF deposits. We also considered a wide range of flow heights, 0.5–33 m based on modelling and observations[23,43]. In a series of numerical simulations, we explore the maximum excess basal pore pressure reached as a function of flow height and fragmentation rates. As expected, thin flows lose excess pore pressure quickly due to diffusion. Slow compaction due to low fragmentation rates lead to excess pore pressure Pg* < 0.2 (Pg* = Pg/normal stress) even for very thick flows (>30 m). However, the generation of excess pore pressure without external gas source or self-fluidisation (Pg* > 0.5) is observed for a wide range of the parameter space investigated (Fig. 3a), with pore pressure reaching as high as 90% of the normal stress.

The granular mixture behaves as a compressible porous media making the basal pore pressure highly dynamic[59]. The pore pressure is controlled by the competing effects of diffusion (sink term) and compaction (source term). This competition is encapsulated by a Deborah number De, which is the ratio of relaxation timescale $t_d$ over the process timescale $t_0$ ('Methods'):

$$\text{De} = t_d/t_0 = \bar{u}_0 \frac{d}{D_c} \qquad (1)$$

The relaxation timescale is the diffusion timescale $= d^2/D_c$, where $d$ is the Sauter mean particle diameter and $D_c$ is the diffusion coefficient. The process timescale $\bar{u}_0/d$ is the compaction timescale, where $u_0$ is the slip velocity (aligned with gravity) between the gas and solid phases (averaged between the two solid phases). The spatial- and time-averaged De for each simulation is plotted against the scaled maximum basal excess pore pressure and shows a power-law scaling. The scaling between Pg* and De holds when the fine fraction size is changed to 70 and 130 microns (Fig. 3b), which yields a Sauter mean diameter spanning 200–360 microns (at the time Pg* is maximum) and overlaps with the typical Sauter mean diameter of BAFs (-125–500 microns)[38].

In granular flows, typically shear (through particle collisions) induces dilation of the mixture rather than compaction, which is known as Reynolds dilatancy[60]. To explore whether dilation could balance the compaction driven by fragmentation of particles, we simulate the evolution of the granular column as it moves along the Merapi slope (Fig. 3c). The fragmentation rate is imposed, and the bed responds to the slope that provides potential energy while the frictional stress dissipates that energy. Our results suggest the fragmentation-induced fluidisation (FIF) that increases the Pg* with distance dominates the first 2–2.1 km of runout, whereas defluidisation dominates the flow beyond that distance. Using a set of 193 simulations to explore the same range of flow heights and fragmentation rates as in the static counter parts (Fig. 3b), we show the maximum Pg* is similar to the static counterpart (Fig. 3a) with only limited effect of the Reynolds dilatancy. Based on a range of fragmentation rates estimated for the 2006 Merapi BAF ('Methods'), we calculate the distance at which the maximum Pg* is reached, which also represents the distance where the flow changes from the FIF regime to a defluidisation regime. It gives distances <2.5 km, which is commensurate with the distance beyond which BAF deposit samples show limited changes in grain size distributions (Figs. 1b and 2g).

Most importantly, the FIF process has a strong influence on the runout of the mixtures, which is predicted to be between 5.5 and 8.5 km for the 2006 BAF (Fig. 3d). The scaled runout is 2.5–3.5, which illustrates the extent to which FIF contributes to the mobility of BAFs and has strong hazard implications. We obtain similar results when changing the slope from Merapi's to an averaged slope from BAF forming volcanoes[61] (see 'Methods'). Additionally, our simulations suggest elutriation of the fine fraction that feeds turbulent ash-cloud surges is largely driven by fragmentation-induced compaction, which matches the phenomenology inferred by recent two-layer depth-averaged modelling of BAFs[10] (Supplementary Note and Fig. 2).

## Impact of the fragmentation-induced fluidisation on PDCs moving across volcanic topography

We use depth-averaged simulations that incorporate a description of the effective shear stress as a function of the excess pore pressure to show the impact of self-fluidisation of BAFs on real 3D volcanic topography. The simulations focus on the June 14, 2006 BAF, which propagated nearly 8 km in the Gendol valley, SE flank of Merapi (Fig. 4a). Since the depth-averaged equations describe the mixture as a single incompressible phase (i.e., no changes of packing concentration), we

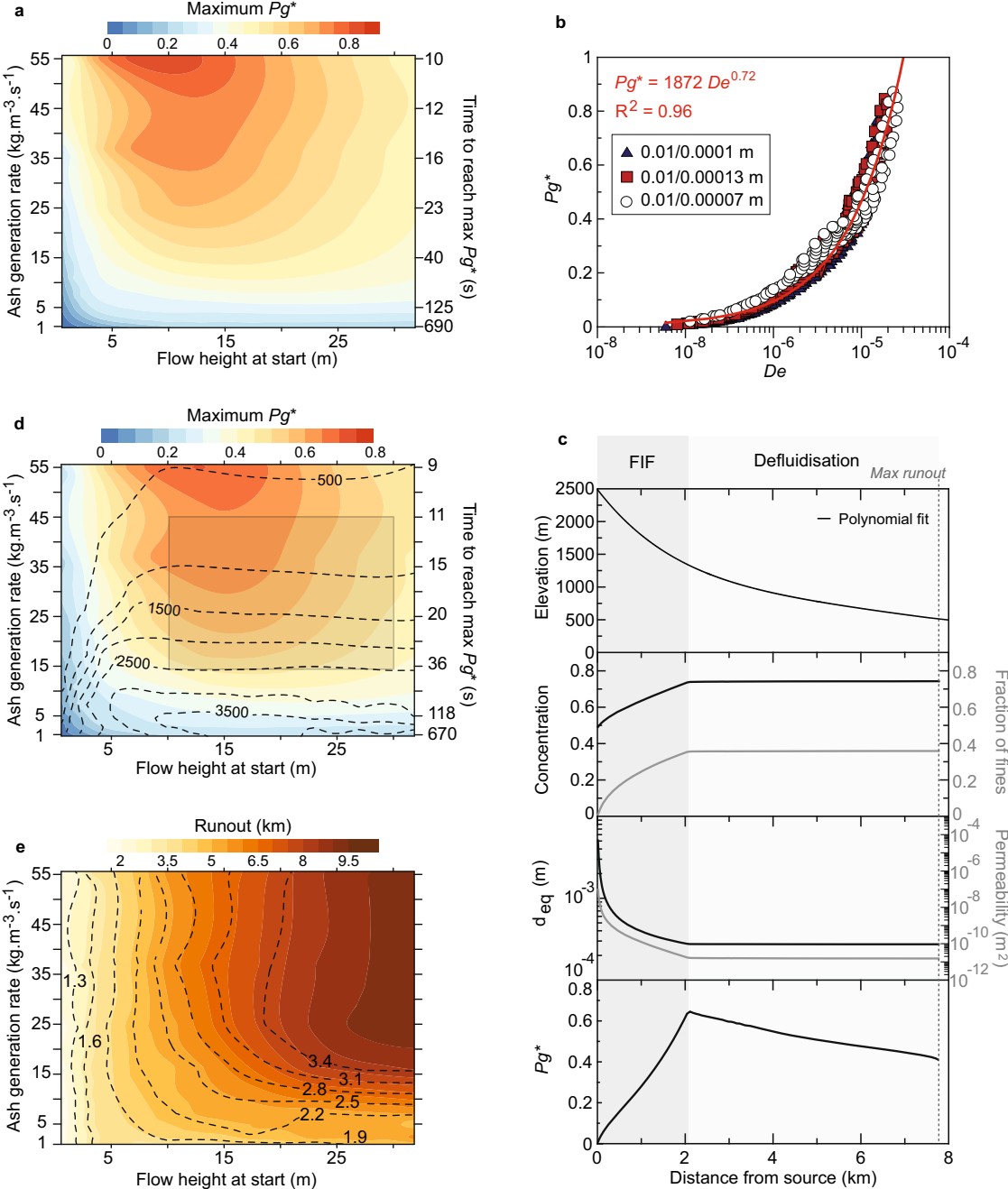

**Fig. 3 | Multiphase modelling of compressible granular media. a** Maximum excess pore pressure generated Pg* (Pg*=pore pressure/lithostatic pressure) as function of flow height and ash generation rates (fragmentation rate) in a static 2D column. **b** Scaling of Pg* with the Deborah (De) number. 193 simulations were run for each bidisperse distribution ($10^{-2}$ m and $10^{-4}$ m; $10^{-2}$ m and $1.3 \times 10^{-4}$ m; $10^{-2}$ m and $7 \times 10^{-5}$ m). A best fit is presented (red line). **c** Numerical simulation results illustrating the evolution of a 18 m thick bed (at $t = 0$ s), undergoing a fragmentation rate of 25 kg m$^{-3}$ s$^{-1}$. The mixture propagates along the Gendol Valley at Merapi volcano (Indonesia). The solid concentration, equivalent diameter, permeability

and scaled excess pore pressure are plotted with distance. **d** Similar to **a** but on the Merapi slope thus allowing shear to develop. The dashed lines are isocontour of the distance at which the maximum excess pore pressure is reached. The grey box represents the range of ash production rate and flow height estimated for the 2006 BAF at Merapi from the most proximal sample collected. **e** Runout of the mixture. The runout is scaled against the runout of the flow with same properties, but no fragmentation occurs (no excess pore pressure). Isocontours of the scaled runouts are shown as dashed lines.

simulate the mixture starting at 2 km from source. The VolcFlow depth-averaged model exists as a two-layer version, describing the basal underflow and dilute ash-cloud. However, since our present work focuses on the dynamics of the dense avalanche, we decided to limit its use to the basal layer. We hypothesise that compaction stops beyond this distance and the mixture' pore pressure is solely driven by diffusion. First, our results show that without FIF (Fig. 4b), the mixture would pile up rapidly because of its high basal friction coefficient of

30° and its propagation on slopes <20°. Instead, the extent of the natural deposit is best reproduced using Pg* = 0.8 and a diffusion coefficient $D_c = 10^{-2}$ m$^2$ s$^{-1}$ (Fig. 4c). We also explore the sensitivity of the results to the initial conditions, showing the diffusion timescale controlled by $D_c$ exerts the strongest control on the flow runout (Fig. 4d and Supplementary Movies 1–13).

The Pg* value used is larger than the predicted value of 0.7 from the multiphase simulations. This is likely related to the non-linearity of

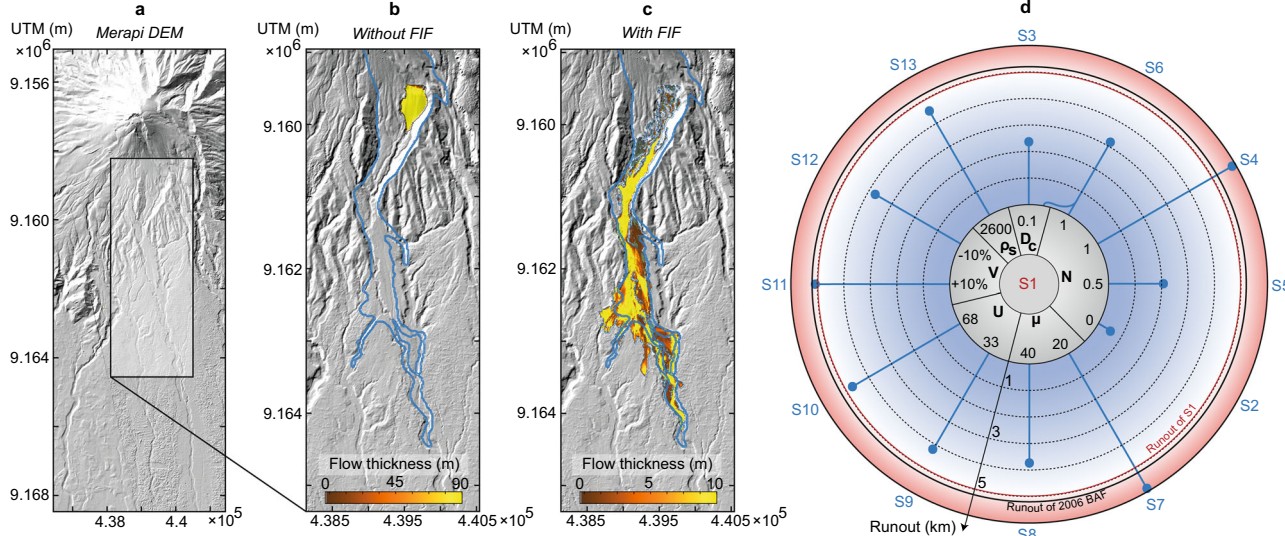

**Fig. 4 | The role of fragmentation-induced fluidisation in enhancing the runout of pyroclastic density currents. a** Digital elevation model (DEM) of Merapi volcano prior to the June 2006 eruption. Depth-averaged model results showing the runout of the BAF mixture across the Merapi topography with and without fragmentation-induced fluidisation. **b, c** VolcFlow model results without (**b**, S2) and with (**c**, S1) fragmentation-induced fluidisation. Parameters used: particle density $\rho_s$ = 2800 kg m$^{-3}$ (Supplementary Fig. 3, diffusion coefficient $D_c$ = 0.01 m$^2$ s$^{-1}$, degree of fluidisation $N$ = 0 (**b**) and $N$ = 0.8 (**c**), basal friction angle $\mu$ = 30°, initial velocity $U$ = 45 m/s, bulk volume $V$ = 6 × 10$^6$ m$^3$. The outline in blue depicts the deposit outline of the June 14 2006 BAF at Merapi that encompasses both the concentrated flow and the ash-cloud surge extent. The colour bar shows the simulated flow thickness in metres. **d** Diagram showing the sensitivity of the flow runout to initial parameters. In each simulation one parameter was modified from the reference case shown in **c**. All UTM coordinates in figures a, b and c are in metres.

fragmentation rates, which could be faster than the estimated upper limit of 45 kg m$^{-3}$ s$^{-1}$, driven for instance by forced-compaction as the mixture propagates across topographical steps on unconfined slopes >30°[30] or due to the fact that friction is simultaneously reduced by acoustic weakening[62]. Additionally, the diffusion coefficient $D_c$ required to match the natural event needs to be on the lower end of the range estimated theoretically for the BAF mixture ($D_c$-0.01–0.1 m$^2$ s$^{-1}$, 'Methods'). While the depth-averaged model assumes BAFs to be incompressible throughout their runout, compaction driven by (1) defluidisation, (2) deceleration of the mixture and (3) continuous comminution could render the flow compressible and further delay the diffusion of excess pore pressure.

Dense PDCs are clearly compressible. This is best illustrated by the newly recognised fragmentation-induced fluidisation (FIF), which helps explain many observations, including how BAFs become fluidised and achieve long runouts (Fig. 5). Fragmentation-induced fluidisation results from the transformation of a coarse-grained granular flow into a bimodal block-and-ash flow with a rapid change of the packing of the mixture and filling of voidage by fines. Rapid compaction induces the generation of excess pore fluid pressure. Consequently, FIF progressively lowers the effective friction coefficient of the volcanic mixture in the first ~2 km where it reaches a minimum and remains low as long as the excess pore pressure can be sustained in the mixture. This explains a long-standing and puzzling observation from inverse modelling that basal friction in BAFs must decrease with distance proximally, thus preventing extreme acceleration of the mixture in the proximal area[63].

Finally, the packing of volcanic mixtures is often poorly constrained, yet flows are sensitive to this property. Hence realistic mixture characterisation should be the focus of future work across experimental, field, and numerical volcanology. Interestingly, the fragmentation-induced fluidisation that leads to the compressible behaviour of block-and-ash flows has never been observed in experiments because the material commonly used (natural volcanic material or analogue material such as glass beads or sand) does not scale with the stress ratio (contact stress/yield stress of material) of natural PDCs.

Thus, compaction timescales are much larger than diffusion timescales in existing experiments[37,64]. Therefore, future work could investigate the dynamics of fragmentation in (fluidised) granular media to help incorporate compressibility in depth-averaged models. Additionally, dome composition (e.g. initial gas content and crystal content can affect its mechanical strength) and collapse mechanism could impact FIF by changing the rate of fragmentation. Thicker flows should create larger fragmentation rates but since deposition in the first kilometres is often negligible, we may not be able to verify such assumption from the deposits of natural events. This work shows there is a need to quantify the rates of particle fragmentation from dome collapse and in-situ particle fragmentation in granular flows and assess the sensitivity to material properties of the particles. Our findings are not only relevant to PDCs, but others natural granular flows immersed in a fluid and where changes of grain-size distribution occur, including debris flows, landslides, rock- and snow-avalanches.

## Methods
### Grain-size analysis of block-and-ash flows
The study of the three PDC deposits was undertaken during multiple field campaigns at Merapi volcano: (1) 2006 and 2007 campaign to study the June 14 2006 block-and-ash flow and (2) 2011 and 2013 campaigns to study the October 26 and November 5 2010 block-and-ash flows. Mapping of the deposits was possible through a combination of field work and analysis of satellite imagery. Detailed stratigraphy of those events and gain-size distributions of the different flow units are available in the literature[41,43,65]. Sampling and laboratory analysis of the grain-size distribution was achieved by the same investigator to ensure the same method was used throughout. In the field, the sample were collected from the middle of the layer to be consistent throughout.

Dynamically significant diameters can be calculated from the grain-size analysis: the D[3,2] = $1/\sum \frac{x_i}{d_i}$, commonly known as the Sauter mean and the volume mean diameter, the D[4,3] = $x_i d_i$. $x_i$ is the weight fraction of particles in the $i$-th sieve, and of mean diameter $d_i = (d_i d_{i+1})^{1/2}$. $d_{i+1}$ is the mesh size of the sieve atop the $i$-th sieve.

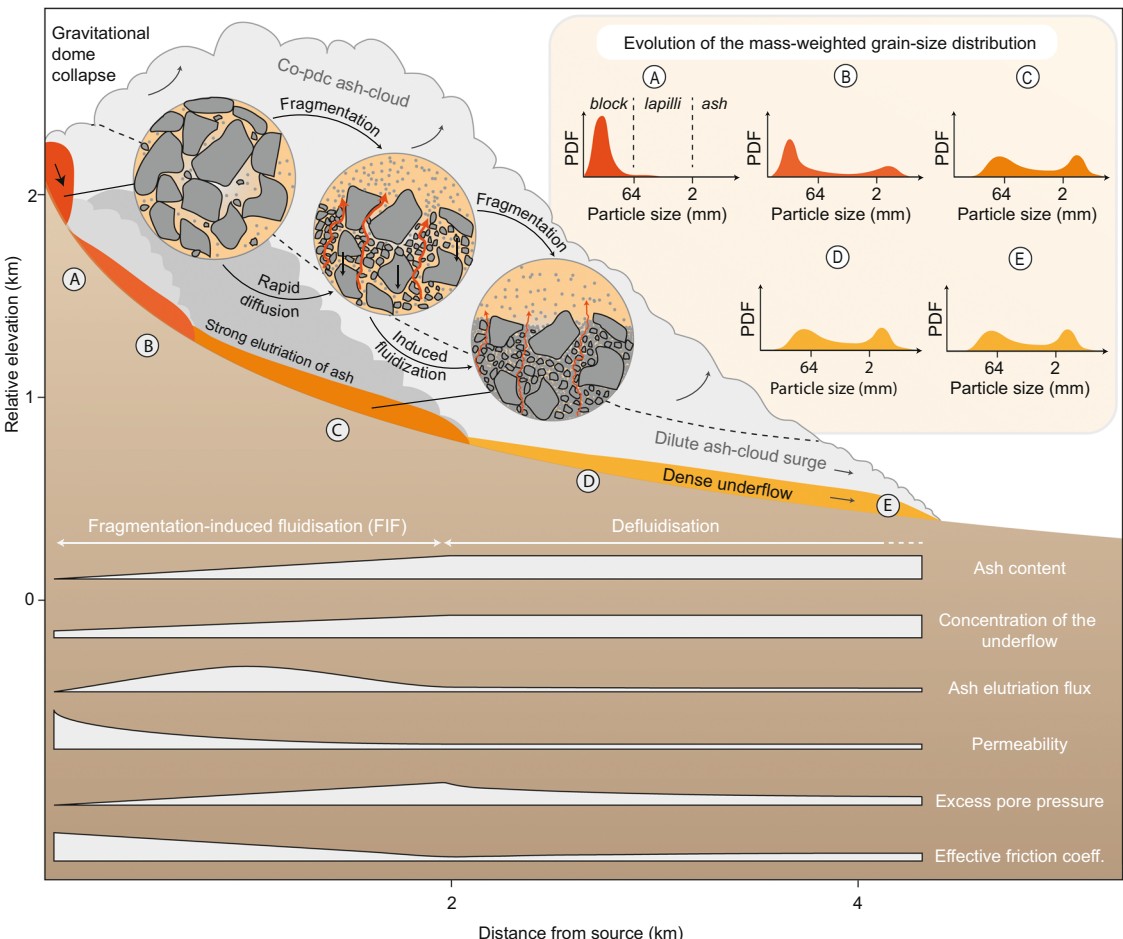

**Fig. 5 | The fragmentation-induced fluidisation of pyroclastic density currents.** Conceptual summary sketch describing the process by which the evolution of the grain-size distribution transported induces compressibility effects in the granular avalanche, which in turn yields partially fluidisation and subsequently lowers the effective friction. The profiles shown are qualitative and not to scale. The grain size distributions are shown in the figure as inserts (A–E) whose locations are depicted in the main schematic drawing with the same labels (A–E).

Estimates of flow heights for the June 14 2006 BAF are based upon results from depth-average modelling of Charbonnier and Gertisser[22] and their observations of superelevation. The ash-generation rates are based on the evidence that (i) 30–40% of the deposit was made of the ash-matrix (<4 mm), (ii) an estimated flow density of 1960–2240 kg m$^{-3}$ (solid density of 2800 kg m$^{-3}$ and particle concentration of 0.75–0.8) and (iii) flow duration to reach the first deposit location of 2.3 km of 20–40 s[22]. These give an estimated ash-generations rates of 15–45 kg m$^{-3}$ s$^{-1}$.

### Laboratory analysis

Block-and-ash flow samples were dry sieved following every phi unit. The clast density analysis of individual 1 phi size fractions was calculated from the ratio of the mass of a bulk sample and the particles' volume. The total porosity is defined as:

$$X_\mathrm{g} = 1 - \frac{m_\mathrm{tot}}{\rho_\mathrm{s} V_\mathrm{tot}} \qquad (2)$$

$\rho_\mathrm{s}$ is the density of the solid, which was obtained by crushing material to a very fine powder, weighing the powder, and measuring the total volume of the sample using a He-pycnometer (Micromeretics AccuPyc II 1340). The mean of five consecutive measurements was used. Equation 2 can be used for other grain-size fractions if we assume that pores in particles are not connected on a length scale similar to

that of the particles. As demonstrated by Colombier et al.[66], the connectivity of the sample is null when the total porosity is <0.2. Following this assumption, we calculated $X_\mathrm{g}$ assuming that $V_\mathrm{tot}$ was provided by the He-pycnometer measurements. Since all our total porosity values are <0.04, it satisfies our initial assumption.

Particle size and shape of the October 2006, 2010 Merapi BAF was studied using a dynamic image analyser (Microtrac PartAn3D) at the University of Oregon. We used 3000 particles in each bin size to compute the average to ensure statistical significance[67].

Analyses of the random-close packing of granular samples was undertaken in the laboratory as follows. The granular mixture was weighed and then poured in a finely graduated cylinder to measure its volume. This provides the bulk density of the mixture, which can be divided by the average density of the particles (from pycnometer analysis) to provide the random-close packing. The method was first validated using glass beads against well-established relationships, and then used with volcanic particles. We found our results to be reproducible and each value is the average of 10 successive experiments with a standard deviation of ~0.05. For each experiment, the material was sieved, weighed, and its bulk volume measured. Gentle tapping (during 30–60 s) on the cylinder was generally used to ensure we find the highest value of packing. To obtain homogeneous bimodal and trimodal grain-size distributions, we pour the mixture layer by layer with a layer thickness ~4 times the coarsest particle diameter.

## Theoretical estimates of random-close packing

Theoretical estimates of the random-close packing were conducted using three approaches. The first one is valid for bidisperse and tridisperse mixtures following the work on spheres by Yu and Standish[54].

$$\varepsilon_{s,mix}^{max} = \left\{ \frac{\varepsilon_s^{max}}{\left(1 - \sum_{j \neq m}^{M}\left(1 - \frac{\varepsilon_m^{max}}{p_{ij}}\right)\frac{cx_i}{X_{ij}}\right)} \right\} \qquad (3)$$

where $\varepsilon_{s,mix}^{max}$ is the maximum packing of the $m$th mixture solid phases. The other terms are defined as follows:

$$cx_i = \frac{\varepsilon_i}{\sum_{j=1}^{M} \varepsilon_j} \qquad (4)$$

$$X_{ij} = \begin{cases} \frac{1-r_{ij}^2}{2-\varepsilon_i^{max}} j < i \\ 1 - \frac{1-r_{ij}^2}{2-\varepsilon_i^{max}} j \geqslant 1 \end{cases} \qquad (5)$$

$$p_{ij} = \begin{cases} \varepsilon_i^{max} + \varepsilon_i^{max}(1 - \varepsilon_i^{max})\left(1 - 2.35 r_{ij} + 1.35 r_{ij}^2\right) & r_{ij}^2 \leq 0.741 \\ \varepsilon_s^{max} & r_{ij}^2 > 0.741 \end{cases} \qquad (6)$$

$$r_{ij} = \begin{cases} \frac{d_{p,i}}{d_{p,j}} i < j \\ \frac{d_{p,j}}{d_{p,i}} j \geqslant i \end{cases} \qquad (7)$$

A second approach was used to calculate the random-close packing of particle size distributions. The spherepack1D algorithm written in C is open-source http://sourceforge.net/projects/spherepack1d/.

When using 2 or 3 particle sizes gives the same results as Yu and Standish (1987), it also allows us to use n-bins describing any grain-size distribution. The code was validated against 3D Discrete Element Method simulations[55], and was used in our work by inputting grain-size data obtained from sieving natural block-and-ash flow samples. A command such as ./spherepack1d.exe -f 13930 -d -p 11 followed by an input of 11 columns and two lines of mean particle size (in the 1 phi bin) and weight fraction can be used to predict the packing of a GSD with 11 narrow bin sizes. The flag -f 13930 allows us to change from packing of spheres with maximum packing of 0.495 to -f 11220 for 0.55.

Estimates of the packing of the initial GSD of the basal underflow was achieved by discretizing a fractal size distribution into 1phi sieve sizes and importing such distribution into Spherepack1D, having taken into account the fact that particles fragmenting due to the fractal distribution of fractures will be non-spherical.

Finally, the packing of polydisperse non-spherical block-and-ash flow mixtures was estimated using a third approach. Recently, Maroof et al.[68] developed an empirical method to calculate the porosity based on the grain-size parameter such as the coefficient of uniformity $C_u = d60/d10$, and the shape factor $\rho = (R + S)/2$, where $R$ is the particle roundness and $S$ is the particle sphericity. The random-close packing is described as follows:

$$\varepsilon_{s,mix}^{max} = 1 - (\rho^{-0.48} C_u^{-0.27} e_{min}^{\circ}) \qquad (8)$$

$e_{min}^{\circ}$ is the minimum voidage for monodisperse spheres = 0.36.

Note the $C_u$ is obtained from the percentiles of the cumulative weight fraction distribution, from fine to coarse size fraction, which is the opposite of what is commonly used in volcanology when one uses the phi scale.

## Multiphase numerical simulations

To gain insights into the impact of compaction by fragmentation of particles and how it impacts the interstitial fluid pressure, we model the mixture using the Eulerian-Eulerian method also known as the two-fluid method (TFM). For simplicity we use the term "numerical model" to refer to a mathematical model solved with numerical methods. 2D numerical simulations were performed with the MFIX open-source code developed by the US Department of Energy's National Energy Technology Laboratory (NETL)[69]. Using this method we solved the mass, momentum equations for the fluid and solid phases. We excluded the energy equations since dense granular mixtures entrain negligible amount of fluid to cool down and cooling over transport distances is negligible for concentrated flows. This is supported by measurements of emplacement temperature of block-and-ash flows[57,58]. The multiphase mixture consists initially of a mixture of air at 773 Kelvin[57,58] and particles of 0.01 m in diameter. The fluid phase is treated as a compressible media using the equation of state of air. The second solid phase of diameter equal to 100 microns is created by fragmentation of the coarse phase. The frictional properties of the mixture is set with a friction coefficient of 0.7 and basal friction coefficient of 0.6. The kinetic stresses are calculated using the Kinetic Theory[70] and the frictional stresses using the Srivastava and Sundaresan model[71], which have been validated[72]. The numerical setup consists of a 2D vertical column that is either static (no slope) or put on a slope. The 2D was used because it was ~10x faster than using 3D with little impact on the results (e.g. max degree of fluidisation varied by 5%), and still allows vertical gradients to develop, such as shear that is not constant with height in granular flow[26].

We mimic the slope by changing the gravity vector in our simulations. Since we wanted to explore how a mixture that fragments would propagate along a volcanic slope, we converted the topographic profile into a slope profile. At initiation, the mixture is placed on the steep slope and responds to that imbalance by shearing. As the velocity develops in the columns, we calculate at each timestep the distance advanced by the column along the volcanic slope. Subsequently, we adapt the gravity vector. This allows us to model the dynamics of the granular columns that fragments as it moves along the slope. As the mixture defluidises and slows down on the shallow slopes, it eventually reaches a full stop.

We imposed rates of fragmentation of the coarse solid phase present spanning $55 \times$ Fraction$_{coarse}$ to $1 \times$ Fraction$_{coarse}$ kg m$^{-3}$ s$^{-1}$, which were introduced in the mass and momentum equations. We also considered a wide range of flow heights, from 0.5 to 33 m, which largely cover natural PDC thicknesses. We did not explore a smaller range that would cover the typical height of experimental flows because we needed to ensure that the grid size was much larger than the maximum particle size (0.01 m) for the two-fluid numerical methods to be valid. The choice of 0.01 m and 0.0001 m was chosen because once blended, the mixture's Sauter mean diameter matches that of natural BAFs, which guarantees the correct coupling between the gas and solid phases through drag. To capture the role of fragmentation on the evolution of the solid concentration in the granular mixture, we incorporated the Yu and Standish[54] description of packing in the model. This impacted the maximum packing value, which is not equivalent to the packing of the mixture, that also depends on the solid pressure.

The input parameters, initial and boundary conditions are summarised in Supplementary Table 1.

The equations describing the numerical models are presented in a technical report by the US Department of Energy[73], and detailed on the friction model are provided in Breard et al.[72].

We performed sensitivity analysis to explore how changes in the particles size, 2D column versus 3D, change of volcanic slope (from

Merapi to a mean volcanic slope) and grid resolution impact the results presented (Supplementary Table 2). These show the robustness of our results and subsequent conclusions drawn from them.

Estimates of flow heights for the June 14 2006 BAF are based upon results from depth-average modelling of Charbonnier and Gertisser[22] and their field observations of super-elevation[43]. The ash-generation rates are based on the evidence that (i) 30–40% of the deposit was made of the ash-matrix (<4 mm), (ii) an estimated flow density of 1960–2240 kg m$^{-3}$ and (iii) flow duration to reach the first deposit location of 2.3 km of 20–40 s[22]. These give an estimated ash-generations rates of 15–45 kg m$^{-3}$ s$^{-1}$.

### Derivation of the scaling parameter De
Assuming a mean grain size with incompressible grains and adiabatic conditions, the compaction and diffusion of porous media is written as follows[74]:

$$\beta \varnothing \frac{\partial P}{\partial t} = \nabla \left[ (1+\beta P)\frac{k}{\mu}\nabla P \right] - (1+\beta P)\nabla \mathbf{u_s} - \beta \varnothing \, \mathbf{u_s}\nabla P \quad (9)$$

$P$ is excess pore pressure, $k$ is the permeability, $\mu$ is the dynamic fluid viscosity, $\beta$ is the fluid compressibility satisfying the equation of state $\rho_f = \rho_0 \,(1+\beta P)$, where $\rho_0$ is the fluid density at a reference hydrostatic level and $\beta = 1/\rho_f \frac{\partial \rho_f}{\partial P}$.

$\varnothing$ is the porosity, $\mathbf{u_s}$ the solid velocity.

$\nabla\cdot$ is the divergence operator and $\nabla$ is the gradient operator.

The first and last terms are the Lagrangian derivatives of the pore pressure, while the second term is the diffusion term and the third term is the forcing due to the difference in grain velocities.

When $\nabla.\mathbf{u_s} < 0$, the bed is compacting. When $\nabla.\mathbf{u_s} > 0$ the bed is dilating.

We can define non-dimensional quantities:
$P = \frac{\bar{P}}{\beta}$; $\mathbf{u_s} = \overline{\mathbf{u_s}}u_0$; $k = \bar{k}k_0$; $t = \bar{t}t_0$; $\nabla = \overline{\nabla_1}\frac{1}{d}$; $\nabla = \frac{\overline{\nabla_2}}{l_k}$

$l_k$ is the diffusion length scale, and $d$ is the effective particle diameter, $l_k = \sqrt{Dt_0}$

We can define the diffusion coefficient $D_c = \frac{k_0}{\beta \varnothing \mu}$ and the timescale of deformation is $t_0 = \frac{d}{u_0}$

We can write Eq. 9 using non-dimensional quantities as:

$$\beta \varnothing \frac{\partial \frac{\bar{P}}{\beta}}{\partial \bar{t}t_0} = \overline{\nabla_1}.\frac{1}{d}\left[ (1+\bar{P})\frac{\bar{k}k_0}{\mu}\frac{\overline{\nabla_2}}{l_k}\frac{\bar{P}}{\beta} \right] - (1+\bar{P})\overline{\nabla_1}.\frac{1}{d}\overline{\mathbf{u_s}}u_0 - \beta \varnothing \overline{\mathbf{u_s}}u_0 \cdot \frac{\overline{\nabla_2}}{l_k}\frac{\bar{P}}{\beta}$$
$$(10)$$

Dividing all terms by $\frac{t_0}{\varnothing}$ and using $D_c$ gives:

$$\frac{\partial \bar{P}}{\partial \bar{t}} = \frac{D_c}{u_0 l_k}\overline{\nabla_1}\cdot\left[ (1+\bar{P})\bar{k}\overline{\nabla_2}\bar{P} \right] - \frac{(1+\bar{P})}{\varnothing}\overline{\nabla_1}.\overline{\mathbf{u_s}} - \frac{d}{l_k}\overline{\mathbf{u_s}}.\overline{\nabla_2}\bar{P} \quad (11)$$

The Deborah number De can be defined as:

$$\frac{D_c}{u_0 l_k} = \sqrt{\mathrm{De}^{-1}} \quad (12)$$

De is the ratio of the relaxation timescale over the process timescale at the scale of the mean grain-size diameter. We here define it as the ratio between the diffusion timescale over the compaction timescale:

De $= \frac{t_d}{t_0}$ where $t_d = \frac{d^2}{D_c}$ and $t_0 = \frac{u_0}{d}$

Note the last term of the Eq. 11: $\frac{d}{l_k}\overline{\mathbf{u_s}}\overline{\nabla_2}\bar{P}$ can be neglected if De $\ll 1$, corresponding to mean particle diameters $>10^{-7}$ m; this is relevant to most volcanic flow applications.

### Depth-averaged modelling
We simulate the flow emplacement over a 8 m resolution DEM using the numerical depth-averaged code VolcFlow (pore pressure version[27]). The code solves mass and momentum balance equations with $x$ and $y$ tangent to the topography, and z is depth-averaged:

$$\frac{\partial h}{\partial t} + \frac{\partial}{\partial x}\left( hu_x \right) + \frac{\partial}{\partial y}\left( hu_y \right) = 0 \quad (13)$$

$$\frac{\partial}{\partial t}\left( hu_x \right) + \frac{\partial}{\partial x}\left( hu_x{}^2 \right) + \frac{\partial}{\partial x}\left( hu_x u_y \right) + \frac{\partial}{\partial x}\left( \frac{1}{2}gh^2\cos\alpha \right) = gh\sin\alpha_x - \frac{T_x}{\rho} \quad (14)$$

$$\frac{\partial}{\partial t}\left( hu_y \right) + \frac{\partial}{\partial y}\left( hu_y{}^2 \right) + \frac{\partial}{\partial y}\left( hu_x u_y \right) + \frac{\partial}{\partial y}\left( \frac{1}{2}gh^2\cos\alpha \right) = gh\sin\alpha_y - \frac{T_y}{\rho} \quad (15)$$

where $\mathbf{u} = [u_x, u_y]$ is the flow velocity, $r$ the mixture density, $h$ its thickness, $\mathbf{T} = [T_x, T_y]$ the dissipative stresses, $g$ the gravity, $\alpha$ the average slope angle, and $\alpha_x$ and $\alpha_y$ the slope angle in $xz$ and $yz$ planes, respectively. Here, the basal gas pore pressure $P_b$ is transported with the rest of the flow and diffused through time, $t$, following (see Supporting Information in Gueugneau et al.[27]):

$$\frac{\partial P_b}{\partial t} + \frac{\partial P_b}{\partial x}u_x + \frac{\partial P_b}{\partial y}u_y = -\left(\frac{\pi}{2}\right)^2 D_c \frac{P_b - P_{atm}}{h^2}. \quad (16)$$

with $P_{atm}$ is the atmospheric pressure and $D_c$ is the diffusion coefficient. The Voellmy–Salm rheology is used to calculate $\mathbf{T}$, combining a turbulent/collisional resistive term with a frictional term that is modified by the basal gas pore pressure $P_b$:

$$\mathbf{T} = \frac{\mathbf{u}}{|\mathbf{u}|}\left[ \left( (P_{atm} + \rho gh \cos\alpha - P_b)\tan\varphi_{bed} + \frac{|\mathbf{u}|^2}{r} \right) + \frac{\rho \mathbf{g}}{\xi}|\mathbf{u}|^2 \right] \quad (17)$$

where $\varphi_{bed}$ is the Coulomb friction coefficient, and $\xi$ the Voellmy coefficient, and $r$ the curvature radius of the topography. This rheology is often used to simulate PDCs[27,75,76]. In our simulations, the fluidisation is generated at the source by $P_b = \delta(\rho gh)$, where $\rho hg$ represents the lithostatic pressure at the base of the flow, and $\delta$ the degree of fluidisation, i.e. the percentage of lithostatic pressure balanced by the gas pore pressure.

## Data availability
All data supporting the findings of this study are available within the article and the Supplementary Information and have been deposited in the open-source Zenodo repository https://doi.org/10.5281/zenodo.7669880. The raw data can be obtained by running the source data files and can be made available by the corresponding author upon reasonable request.

## Code availability
The MFIX code is open-source and available at https://mfix.netl.doe.gov/. The custom changes to MFIX subroutines are presented in the following repository https://doi.org/10.5281/zenodo.7669880. SpherePack1D is available at the following link https://sourceforge.net/projects/spherepack1d/files/ and the VolcFlow code can be found at https://lmv.uca.fr/volcflow/.

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

## Acknowledgements

E.C.P.B. was supported by UKRI with the NERC-IRF (NE/V014242/1). J.D. was supported NSF EAR 1852569 and EAR 1650382. S.J.C. and V.G. acknowledge funding from the National Science Foundation (NSF) CAREER project #17511905. The present work was conceptualised during the 2019 Cooperative Institute for Dynamic Earth Research (CIDER) summer programme in Berkeley, CA, USA, supported by NSFEAR1135452.

## Author contributions
E.C.P.B. designed the study, conducted the laboratory experiments, ran the multiphase flow numerical simulations and conducted the analytical packing analysis, and wrote the first draft of the manuscript, which was revised by J.D., S.C., V.G., T.G. and B.W. S.C. conducted the fieldwork and the grain-size analyses. V.G. ran the VolcFlow simulations.

## Competing interests
The authors declare no competing interests.
