## [Peer Review File · Nature Communications]

The fragmentation-induced fluidisation of pyroclastic density currentsREVIEWER COMMENTS

Reviewer #1 (Remarks to the Author):

This paper explores the important and poorly explored controls on fluidisation and mobility of block and ash flow PDCs. Using field data to validate, a numerical model is applied to look at how fragmentation rates impact the grainsize distribution and resulting fluidisation potential which controls changes in mobility at different points in the flow. This is supported with robust presentation of data, and useful supplementary materials.

The fluidisation of BAFs is something of an elephant-in-the-room of PDC modelling, and something I was in the preparatory stages of approaching myself. This work moves the discussion forward substantially, and I'm very pleased to see it carried out so coherently.

My only over-arching concern with the manuscript as it stands is one of terminology. The language and definitions surrounding the different current types and parts of currents get quite mixed quite quickly, and I'm concerned there is a creeping useage of some older terms which carry some inherent implications and inferences which might not apply to what the authors are trying to communicate. There is sometimes a blurring of process/observation/description in the terms being used. The descriptions in Para 1 are presented as being specific to BAF, but obviously are applied more widely too, yet it's not clear where the lines for BAF-specific description lie. For example, how do "pumice flows" (line 51) relate to the definitions on lines 39-40 – this seems to set up a dichotomy where PDCs are either BAFs or pumice flows. Some of this terminology is no longer well used in the literature. I would make the case that "pumice flow" carries a number of implications which cloud the water, and that "column collapse PDC" would be better. Similarly, "dense" "concentrated", "turbulent", "dilute" etc carry descriptive power without blurring the lines into what is and isn't surge, underflow, pyroclastic flow, etc. The descriptions would also be aided by more quantification, or at least qualification (e.g. – what is meant by "course" in line 52, or "fines" in line 99).

I have a few line-by-line comments below which I hope may be of use.

Pete Rowley

Line by line comments

Line 36 – The proximal tephra definition is new to me, although I am in complete agreement that the style of current is the same – it would be useful to cite where else this has been used (if it has), or to spend a little more time explaining what aspects of the current/source/load define it as a BAF.

Line 38-39 – Mixture of tenses being used – could be more clearly phrased.

Line 60 – give the use of matrix in the following line it would be more correct to talk about BAF deposits rather than BAFs.

Line 65 – space between fragmentation and induced

Line 64-70 – Somewhere in here it might be useful to highlight the existing work on grain size dependence of fluidization / pore pressure diffusion.

Paragraph starting line 78 – this section should more clearly delineate what are model inputs/assumptions, and what are description of the natural data. The information is in here, but it is not as clearly distinguished as perhaps it should be.

Line 78 – BAF, not BAFs

Line 81 – This arbitrary approach is fine, but it might be instructive to identify where the GSD-equivalent locations are for different scales of current. More of a future piece of work than for here though.

Line 85 – self-similar

Line 90 - dome collapse

Line 95 – few changes or little change.

Line 105 – Terminology – if BAF are defined by their start conditions, can they transform to ash flows? What is the definition of an ash flow being used here? Given the hard cutoffs in grain size definitions, and how little that 2 mm ash definition is likely to control overall process (1.9 mm ash vs 2.1 mm lapilli are going to behave almost identically) I would advise staying away from grain size-derived current descriptions. BAFs are a special case in many ways (and perhaps better terminology is needed for them as a community), so I would steer clear from blurring those descriptive lines.

117-118 – matrix is specifically a term for a deposit. If you specifically wish to discuss the current I would perhaps stick to “fines fraction” or similar.

216-217 – Given the explicit mention of BAFs existing as 2 layer flows at the beginning of the paper, and the anticipated expulsion of fines into the dilute layer (e.g. Kelfound & Gueugneau 2022) a paragraph addressing the validity of the depth averaging assumption would be useful.

248-250 – this statement feels like it belongs more in the summary/conclusions section

Figures – these are beautifully constructed and presented. My only concerns are (1) the caption at the top of 2e, where “little change in concentration” might be better replaced by “little change in packing”, and (2) you use the term co-ignimbrite ash cloud in figure 5 which is fine, except it doesn’t tally with any

previous description or definition using those terms. (3) in Figure 5 the A-E grainsize illustrations are great, but it might be better to try to overlay them so that the sometimes subtle differences between them can be seen. (4) in Figure 5 would it be possible/useful to add a curve to the lower charts showing ash generation rate to better highlight the balance between ash elutriation, ash content, etc?

Line 377 – Packing is presented in terms of fractions but you're stating your SD in terms of percent – it might be useful to bring one in line with the other.

Line 378 – an indication of the duration of tapping would be useful.

Line 411- text “using a third approach” is unnecessary.

Reviewer #2 (Remarks to the Author):

The paper represents an original contribution to elucidate the mobility of block-and-ash flow (BAF), one of the deadliest volcanic phenomena. The authors are proposing the fragmentation-induced fluidization (FIF) process as the main mechanism that lets a permeable granular flow develop fluidization through frictional fragmentation of grains, which in turn promotes the increase in fine fraction, decreases permeability and increases granular-packing and pore pressures. This process occurs in the proximal reaches, after which the flow continues to emplace due to defluidisations. The model is based on data obtained from three eruptive episodes at Merapi volcano, numerical models and theoretical assumption. The implications of the model here proposed are multiple, one of which is hazard assessment. The paper is very well organized, clear and plenty justified. I only have a few comments. For example, the model assumes that the eruptive mechanism related to the initiation of a block-and-ash flow does not control the FIF process. For the three analyzed cases, the FIF process occurs from the source area up to 2 km: 1) BAF can originate from different initial conditions, with a variable amount of gas (ie. Fast [hot] dome growing, boiling-over). Can a different initial gas content affect the FIF processes? 2) The events observed at Merapi had different maximum runout, but apparently, for all of them, the FIF dominated for the first 2 km on a slope $> 35^\circ$. So, which are the factors that control the distance over which the FIF takes place? The initial volume of the mass or the volcano slope or none? A relation of 2.5-3.5 between the total BAF runout and the distance over which FIF occurs is here proposed: Can this proportion be applied to other volcanoes? 3) Can a larger amount of gas in the initial rock mass retard the FIF process (i.e. >particles elutriation)?

Line 89. Should be Fig. 1d.

Reviewer #3 (Remarks to the Author):

This is an outstanding and fundamentally important contribution to our understanding of concentrated pyroclastic currents, specifically block-and-ash flows that are the most frequent of these currents. The work shows that reduction in porosity due to fragmentation of clasts along flow length works to keep pore pressures relatively high, which in turn reduce friction and promote fluid-like behavior of the gas-particle mixtures. The work is extremely well presented, and the integration of field data and different modeling approaches (one that is used mainly for research, and another that is also used for hazard assessments) is excellent. My comments on the marked-up pdf are minor clarifications, I advise accepting the paper after minor revisions addressing my comments/edits. I will note that more information on modeling parameters should be presented - enough that an independent (qualified) researcher would be able to reproduce the results. This is in reference to Table 1 in extended data. In the multiphase model description (Methods) I can understand why the governing equations are not presented (space), but please make sure that a reference is provided that clearly lays them out exactly as used in this paper (including granular temperature and kinetic theory).

Excellent work, congratulations to the authors, and I look forward to seeing this published. -Greg Valentine

Please find below our point-by-point response and detailed changes in response to the three reviewers' detailed comments:

Reviewer 1:

C1: My only over-arching concern with the manuscript as it stands is one of terminology. The language and definitions surrounding the different current types and parts of currents get quite mixed quite quickly, and I'm concerned there is a creeping useage of some older terms which carry some inherent implications and inferences which might not apply to what the authors are trying to communicate. There is sometimes a blurring of process/observation/description in the terms being used. The descriptions in Para 1 are presented as being specific to BAF, but obviously are applied more widely too, yet it's not clear where the lines for BAF-specific description lie. For example, how do "pumice flows" (line 51) relate to the definitions on lines 39-40 – this seems to set up a dichotomy where PDCs are either BAFs or pumice flows. Some of this terminology is no longer well used in the literature. I would make the case that "pumice flow" carries a number of implications which cloud the water, and that "column collapse PDC" would be better. Similarly, "dense" "concentrated", "turbulent", "dilute" etc carry descriptive power without blurring the lines into what is and isn't surge, underflow, pyroclastic flow, etc. The descriptions would also be aided by more quantification, or at least qualification (e.g. – what is meant by "course" in line 52, or "fines" in line 99).

Reply: We clarified the distinction between block-and-ash flow, and block-and-ash flow deposit in the introduction line 64 and in the methodology. We have changed the term "pumice flow" by "column collapse PDC" at lines 51 and 58, as we agree it may carry more weight than intended. Our manuscript jumps straight into block-and-ash flows as they are the type of PDCs affected by fragmentation-induced fluidisation. Column collapse PDCs acquire their GSD mostly from conduit fragmentation and secondary fragmentation in the conduit, as described by Dufek, J., et al. (2012). "Granular disruption during explosive volcanic eruptions." *Nature Geoscience* 5(8): 561-564. We have included a definition for coarse blocks (>64 mm) and fines-rich (<4mm) on line 61-62.

C2: Line 36 – The proximal tephra definition is new to me, although I am in complete agreement that the style of current is the same – it would be useful to cite where else this has been used (if it has), or to spend a little more time explaining what aspects of the current/source/load define it as a BAF.

Reply: We changed the term "proximal" by "perched", as it was used in the reference we cite on Volcan de Fuego PDCs.

Reference: Risica, G., Rosi, M., Pistolesi, M., Speranza, F. & Branney, M. J. Deposit-Derived Block-and-Ash Flows: The Hazard Posed by Perched Temporary Tephra Accumulations on Volcanoes; 2018 Fuego Disaster, Guatemala. *J. Geophys. Res. Solid Earth* 127, e2021JB023699, doi:<https://doi.org/10.1029/2021JB023699> (2022).

C3: Line 38-39 – Mixture of tenses being used – could be more clearly phrased.

Reply: We rephrased it as: "Observations and depositional evidence suggest these flows are characterized by a concentrated basal avalanche (underflow) overlain by a dilute turbulent ash-cloud (ash-cloud surge)".

C4: Line 60 – give the use of matrix in the following line it would be more correct to talk about BAF deposits rather than BAFs.

Reply: We added “deposits” after “BAF”, as suggested.

C5: Line 65 – space between fragmentation and induced

Reply: We believe this is a compound adjective, in which case the hyphen used is appropriate.

C6: Line 64-70 – Somewhere in here it might be useful to highlight the existing work on grainsize dependence of fluidization / pore pressure diffusion. Paragraph starting line 78 – this section should more clearly delineate what are model inputs/assumptions, and what are description of the natural data. The information is in here, but it is not as clearly distinguished as perhaps it should be.

Reply: We agree and it now reads:

As demonstrated in experimental studies, the permeability and pore-pressure diffusion timescales are controlled by the polydispersity, abundance of fines, and porosity. A key component of the fragmentation-induced fluidisation process is a reduction in the void space through the production of fine particles via cascading fragmentation³⁷⁻³⁹.

C7: Line 78 – BAF, not BAFs

Reply: Corrected.

C8: Line 81 – This arbitrary approach is fine, but it might be instructive to identify where the GSD-equivalent locations are for different scales of current. More of a future piece of work than for here though.

Reply: We agree that this will be an important to quantify the sensitivity both scale and material properties in future work. We have added this in lines 326-330.

C9: Line 85 – self-similar

Reply: It is already written as self-similar.

C10: Line 90 - dome collapse

Reply:

C11: Line 95 – few changes or little change.

Reply: we use “little change”.

C12: Line 105 – Terminology – if BAF are defined by their start conditions, can they transform to ash flows? What is the definition of an ash flow being used here? Given the hard cutoffs in grainsize definitions, and how little that 2 mm ash definition is likely to control overall process (1.9 mm ash vs 2.1 mm lapilli are going to behave almost identically) I would advise staying away from grainsize-derived current descriptions. BAFs are a special case in many ways (and

perhaps better terminology is needed for them as a community), so I would steer clear from blurring those descriptive lines.

Reply: We changed it to: “As a result, this self-limiting process prevents the transformation of (bimodal) block-and-ash flows into (unimodal) ash-dominated flows.”

C13: 117-118 – matrix is specifically a term for a deposit. If you specifically wish to discuss the current I would perhaps stick to “fines fraction” or similar.

Reply: Agreed. We changed it to “fines fraction”.

C13: 216-217 – Given the explicit mention of BAFs existing as 2 layer flows at the beginning of the paper, and the anticipated expulsion of fines into the dilute layer (e.g. Kelfound & Gueugneau 2022) a paragraph addressing the validity of the depth averaging assumption would be useful.

Reply: Agreed. We added:

“The VolcFlow depth-averaged model exists as a two-layer version, describing the basal underflow and dilute ash-cloud. However, since our present work focuses on the dynamics of the dense avalanche, we decided to limit its use to the basal layer.”

C14: 248-250 – this statement feels like it belongs more in the summary/conclusions section

Reply: This statement is the last paragraph of the manuscript, which summarizes the impact of our study.

C15: Figures – these are beautifully constructed and presented. My only concerns are (1) the caption at the top of 2e, where “little change in concentration” might be better replaced by “little change in packing”, and (2) you use the term co-ignimbrite ash cloud in figure 5 which is fine, except it doesn’t tally with any previous description or definition using those terms. (3) in Figure 5 the A-E grainsize illustrations are great, but it might be better to try to overlay them so that the sometimes subtle differences between them can be seen. (4) in Figure 5 would it be possible/useful to add a curve to the lower charts showing ash generation rate to better highlight the balance between ash elutriation, ash content, etc?

Reply: We made the corrections suggested to figure 2. The differences in Figure 5 between GSDs are clear. They are subtle between D and E because they reflect the lack of change of GSD with distance (as observed in natural events). We appreciate the suggestion but feel it would be a repetition of the ash content presented.

C16: Line 377 – Packing is presented in terms of fractions but you’re stating your SD in terms of percent – it might be useful to bring one in line with the other.

Reply: We write it as “0.05”.

C17: Line 378 – an indication of the duration of tapping would be useful.

Line 411- text “using a third approach” is unnecessary.

Reply: It now reads: “Gentle tapping (during 30-60 s) on the cylinder was generally used to ensure we find the highest value of packing.”

Reviewer #2 (Remarks to the Author):

C1: The paper represents an original contribution to elucidate the mobility of block-and-ash flow (BAF), one of the deadliest volcanic phenomena. The authors are proposing the fragmentation-induced fluidization (FIF) process as the main mechanism that lets a permeable granular flow develop fluidization through frictional fragmentation of grains, which in turn promotes the increase in fine fraction, decreases permeability and increases granular-packing and pore pressures. This process occurs in the proximal reaches, after which the flow continues to emplace due to defluidisations. The model is based on data obtained from three eruptive episodes at Merapi volcano, numerical models and theoretical assumption. The implications of the model here proposed are multiple, one of which is hazard assessment. The paper is very well organized, clear and plenty justified. I only have a few comments. For example, the model assumes that the eruptive mechanism related to the initiation of a block-and-ash flow does not control the FIF process. For the three analyzed cases, the FIF process occurs from the source area up to 2 km: 1) BAF can originate from different initial conditions, with a variable amount of gas (ie. Fast [hot] dome growing, boiling-over). Can a different initial gas content affect the FIF processes?

Reply: This is a great point. We have added in the conclusion a sentence to motivate future work in this direction:

Additionally, dome composition (e.g. initial gas content and crystal content can affect its mechanical strength) and collapse mechanism could impact FIF by changing the rate of fragmentation. Thicker flows should create larger fragmentation rates but since deposition in the first kilometres is often negligible, we may not be able to verify such assumption from the deposits of natural events. This work shows there is a need to quantify the rates of particle fragmentation from dome collapse and in-situ particle fragmentation in granular flows and assess the sensitivity to material properties of the particles.

C2: 2) The events observed at Merapi had different maximum runout, but apparently, for all of them, the FIF dominated for the first 2 km on a slope $> 35^\circ$. So, which are the factors that control the distance over which the FIF takes place? The initial volume of the mass or the volcano slope or none?

Reply: Great question. These would be good avenues to pursue and touches on rock mechanics. Due to lack of existing mathematical model describing fragmentation for a continuum framework, we explore imposed rates of fragmentation. Therefore, we cannot answer the question using our current results. We expect that volume, or rather thickness, would play an important role in fragmentation rates as the fragmentation of particles requires the contact stress to exceed the yield strength of the material. Thicker flows yield larger contact forces that are transferred through force chains for instance.

The slope in the Gondol valley drops below 35 degrees at 500 m or so, and at 2 km is at 18 degrees. This is based on the 2 m DEM.

C3: A relation of 2.5-3.5 between the total BAF runout and the distance over which FIF occurs is here proposed: Can this proportion be applied to other volcanoes?

Reply: BAF-forming volcanoes have quite a self-similar slope= $f(\text{distance})$ profile, as shown by Kelfoun 2011. We have ran the same simulations presented in Fig.3 on the mean volcanic slope Kelfoun 2011 defined and noticed little change to our results. This is presented in the Extended Data Table 2.

Therefore, we infer the proportion could be applied to other volcanoes.

We added at line 210: “We obtain similar results when changing the slope from Merapi’s to an averaged slope from BAF forming volcanoes⁶⁰ (See Methods).”

Reference: K. Kelfoun, Suitability of simple rheological laws for the numerical simulation of dense pyroclastic flows and long-runout volcanic avalanches, Journal of Geophysical Research 2011 Vol. 116 Issue B8

C4: 3) Can a larger amount of gas in the initial rock mass retard the FIF process (i.e. >particles elutriation)?

Reply: This is a possibility. The trade-off is that the mixture needs to fragment to develop a low permeability, otherwise most of the excess pore pressure will diffuse very rapidly (<10s) even if the flow is tens of meters in thickness. The current summary addresses the need to study fragmentation in detail, thus answering this comment.

C5: Line 89. Should be Fig. 1d.

Reply: This was corrected.

Reviewer #3 (Remarks to the Author):

C1: This is an outstanding and fundamentally important contribution to our understanding of concentrated pyroclastic currents, specifically block-and-ash flows that are the most frequent of these currents. The work shows that reduction in porosity due to fragmentation of clasts along flow length works to keep pore pressures relatively high, which in turn reduce friction and promote fluid-like behavior of the gas-particle mixtures. The work is extremely well presented, and the integration of field data and different modeling approaches (one that is used mainly for research, and another that is also used for hazard assessments) is excellent. My comments on the marked-up pdf are minor clarifications, I advise accepting the paper after minor revisions addressing my comments/edits. I will note that more information on modeling parameters should be presented - enough that an independent (qualified) researcher would be able to reproduce the results. This is in reference to Table 1 in extended data. In the multiphase model description (Methods) I can understand why the governing equations are not presented (space), but please make sure that a reference is provided that clearly lays them out exactly as used in this paper (including granular temperature and kinetic theory).

Excellent work, congratulations to the authors, and I look forward to seeing this published. -
Greg Valentine

Reply: We have added in the Methods section: “The equations describing the numerical models are presented in a technical report by the US Department of Energy⁷², and detailed on the friction model are provided in Breard, et al. ⁷¹”. In addition, we share an example of MFIX input files in the updated Zenodo repository. This will enable an independent qualified user to reproduce the results.

C2: Is it reasonable to compare GSDs without data on a significant portion of the distributions? Does the block size change along flow? To me it looks like they essentially don't change even in proximal...?

Reply: Ideally, we would have total GSDs, however it could not be achieved there. However, it will be of value to measure the total GSD in future studies to better constrain the FIF process. By analysing the same size range of the GSD, even if we exclude the block sizes, we can tell whether the FIF process (that generates ash size fractions) was active or not.

Most of the sampling was achieved towards the transition from proximal to medial portions of the deposits.

C3: is there a chance this could be read as "to the 49th power"?

Reply: This is in reference to the citation format. We have placed the citation after the dot to avoid any confusion.

C4: This makes sense. I think it would be good to refer back to the GSDs in Fig. 1 and mention that what is labeled as "proximal" in 1b is already beyond this transition region. Otherwise it can be a bit confusing, as seen in my previous comment on fig 1b.

Reply: The samples “proximal” are at a distance $<1/3^{\text{rd}}$ of the final runout but close to the transition. We have thus included a mention in the Fig.1 caption as suggested.

“Note that samples in the “proximal” region were collected close to the proximal/media transition.”

C5: In Extended Data Table 1 please add information on grid resolution and other information that would be necessary for a qualified independent researcher to reproduce the work without guessing on parameters such as equations of state, particle properties, etc.

Reply: In the Extended Data Table 1, we included the grid resolution information and that air is the gas species that is used in the Eq. of state. We included a copy of the MFIX model files in the revised submission.

C6: In the caption for 4c, correct final sentence to read: "...the simulated flow thickness in meters."

Figure 4 is important - the images and labels are very small. Consider reformatting to be larger so they are easily read on a pdf of normal (print) size.

Reply: We agree and have corrected the caption. We also made the text larger in figure 4.

REVIEWERS' COMMENTS

Reviewer #2 (Remarks to the Author):

I have reviewed the new version of the manuscript and I consider that the authors answered all my questions. Therefore, I highly recommend the publication of the article, which represents a very original contribution about BAFs mobility.

Lucia

Reviewer #2 (Remarks to the Author):

C1: I have reviewed the new version of the manuscript and I consider that the authors answered all my questions. Therefore, I highly recommend the publication of the article, which represents a very original contribution about BAFs mobility.

Reply: Thank you, we are glad to hear.